# A Brief Report of Hotel Collapse Causing Casualties in Suzhou, China

**Yu-Lin Chen [1], Pierre Guy Atangana Njock [2],\* and Lin-Shuang Zhao [2],\***

[1] Department of Civil Engineering, School of Naval Architecture, Ocean and Civil Engineering, Shanghai Jiao Tong University, Shanghai 200240, China; chenyvl@sjtu.edu.cn

[2] MOE Key Laboratory of Intelligence Manufacturing Technology, Department of Civil and Environmental Engineering, College of Engineering, Shantou University, Shantou 515063, China

\* Correspondence: njock@stu.edu.cn (P.G.A.N.); lshzhao@stu.edu.cn (L.-S.Z.);
Tel.: +86-754-8650-4551 (P.G.A.N.)

**Abstract:** The collapse of a 30-year-old hotel building in Suzhou, Jiangsu Province on 12 July 2021 raised legitimate questions about the identification of old buildings' condition and risks stemming from remedial operations. This short communication reports and investigates the causes of this accident, which led to 17 deaths and 5 injuries. Subsequently, it describes the rescue actions undertaken, including logistic means, operational strategies, and procedure sequencing. The causes of the accident were attributed to: (i) the poor quality and fragility of the building, (ii) illegal renovations and extensions, as well as (iii) the laxism of relevant departments that failed to timely check the risk level of the building before these renovations. Thanks to efficient organization and management, the rescue operations were completed within 42 h. Based on this preliminary analysis, some recommendations are proposed to prevent similar incidents in the future.

**Keywords:** accident; building collapses; rescue measures; building renovation; recommendations

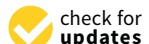



## 1. Introduction

The design, construction, and management of urban buildings have become critical concerns over the years, given the rather frequent occurrence of accidental events. Despite the continuous development of new technologies [1,2], many urban buildings still fail to achieve their primary function, i.e., providing life safety and/or ensuring the physical integrity of residents. This situation is often manifested in several building collapse accidents in many cities around the world. Particularly, China has experienced recurrent building collapse accidents in the past few years, as shown in Table 1 [3]. Generally speaking, the occurrence of building collapse accidents can be ascribed to three main reasons, including the age of buildings, unreasonable structural design, and natural disasters [4–9]. Such events usually have disastrous consequences, amongst which human causalities remain the most abysmal. In response to that, several studies on the issue of building collapses have been performed [10–12], and many methods to prevent the occurrence of this problem have been proposed [13–22]. For instance, Li et al. [13] proposed a method to identify collapsed buildings using remote sensing in earthquake-prone areas and provided general recommendations for preventing building collapses. Ongbali et al. [14] evaluated the latest building structural health monitoring approaches and supplied some promising perspectives for modern buildings. Pearson et al. [15] exposed serious defects in design and construction following a high-rise building collapse accident in London. Daniel et al. [16] investigated a collapse accident of a frame structure under construction and put forward many suggestions for the safety of frame structure buildings. Friedman et al. [17] gave advice to the design and construction of cage-frame structure buildings, taking the collapse of Darlington Apartments as example. However, the methods proposed in recent research are not suitable for masonry-concrete structure buildings, and recent building collapsing

accidents in China show that the majority of collapsed buildings are masonry-concrete structure buildings.

**Table 1.** Building collapses in recent years in China (Summarized from [3]).

| Date | Collapse Events | Cause | Casualty |
|---|---|---|---|
| 29 August 2020 | A restaurant in Xiangfen County, Shanxi Province | Illegal building extension | 29 killed and 29 injured |
| 4 August 2020 | A warehouse in Harbin | Illegal building renovations | 9 killed |
| 7 March 2020 | Xinjia Hotel, Quanzhou City, Fujian Province | Illegal building extension | 29 killed |
| 20 May 2019 | A bar in Baise City, Guangxi Province | Illegal building renovations | 6 killed and 87 injured |
| 16 May 2019 | A factory in Shanghai | Illegal construction | 12 killed and 13 injured |

Another restriction of current research is that existing research on the collapsing of a masonry-concrete structure mainly focuses on the stability analysis under load in special circumstances [23,24]. For instance, Yang et al. [23] summarized the typical failure patterns of brick–concrete structure buildings under earthquake loads in areas with different seismic intensity. Wang [24] analyzed the characteristics and causes of the damage of brick–concrete buildings after an earthquake and proposed some suggestions and measures for the reinforcement of brick–concrete buildings. There are few studies on the stability of structure itself. It is difficult to analyze the mechanics and check the stability of a brick–concrete structure due to its early construction age. Additionally, the above literature [13–24] rarely analyzes the management problems, so more attention should be paid to management problems. Given the continuous occurrence of building collapses, it is vital and urgent to contribute to the knowledge of the field via providing punctual insights from new collapse cases. Additionally, direct causes of collapse, management problems, and corresponding remedies need to be proposed.

This paper presents the preliminary investigation into the cause of a recent hotel collapse in Wujiang, Suzhou, which resulted in 17 deaths and serious injuries to 5 people. The objective of this study is to contribute to the understanding of old building collapses accidents and supply effective recommendations. The background of this accidental event is first introduced, and then rescue operations are presented. Subsequently, a discussion on direct causes of collapse, management problems, as well as some measures to avoid similar accidents are proposed.

## 2. Background

Figure 1 shows the location of the collapsed hotel as well as its appearance before the accident. This accidental event took place in the Wujiang district of Suzhou, Jiangsu Province and involved the "Four seasons Kaiyuan Hotel". The hotel consisted of two buildings relatively close to each other: a main building and an auxiliary building (Figure 1b). The main building (four-story) was intended to provide accommodation, catering, and other hotel services, while the auxiliary building (four-story) was mainly designed to host entertainment facilities such as guest rooms or chess and card rooms. In terms of structural design, the hotel used to adopt a reinforced mixed brick wall structure with a steel roof.

In 2018, the "Four seasons Kaiyuan Hotel" was renovated to adapt to the owner's business strategies. According to media reports, when the owner acquired the hotel in 2018, the main and auxiliary buildings were structurally connected by a wall that already revealed some cracks. In addition, several guests who stayed at the hotel few days before the collapse said that they could hear distinctive cracking noises in their rooms. Unfortunately, these noises were warning signs that ultimately evolved into its breakdown. It should be noted, however, that the collapse did not affect the main building of the four-story hotel, but its auxiliary building.

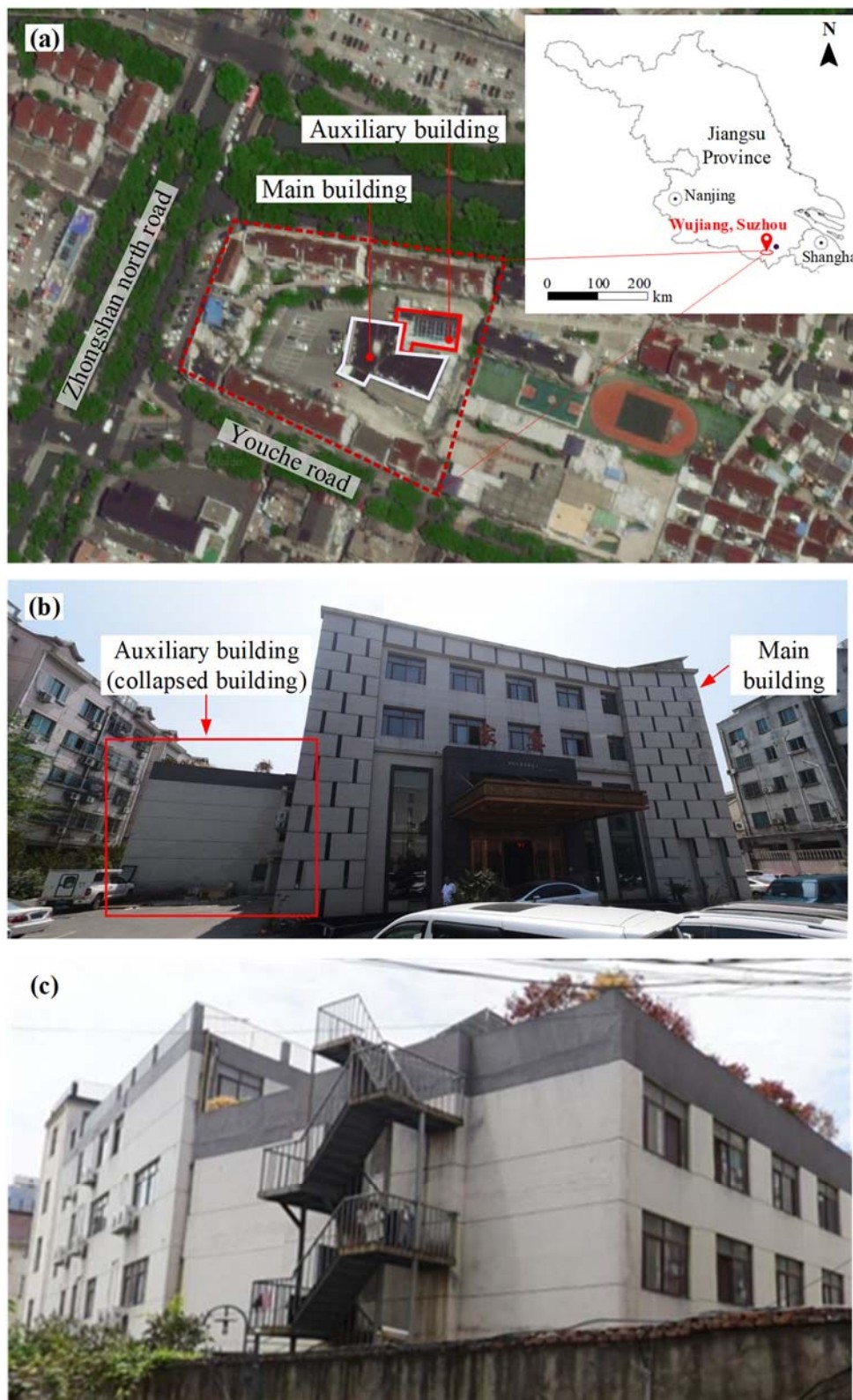

**Figure 1.** (**a**) Location of the building collapse site; (**b**) a view of hotel facade; (**c**) a view of the auxiliary building before the accident (Source: https://map.baidu.com/@13431247.69,3632620.17,19z, accessed on 13 July 2021).

### 3. Building Collapse and Rescue Operations

*3.1. Investigation of the Collapse Accident*

At about 15:33 on 12 July 2021, the auxiliary building of the Four Seasons Kaiquan Hotel collapsed in Wujiang District, Suzhou. Nearby witnesses reported that the collapse happened in a "split second", causing a loud noise and a cloud of dust. Figure 2 shows an aerial view of the buildings before and after the accident. As can be seen from the picture, the structure of the main building remained basically intact, while the collapsed auxiliary building was completely destroyed. Moreover, according to the neighboring residents and customers who stayed at the hotel few days before the accident, the hotel underwent renovation works, during which the shaking of the floor could be clearly felt.

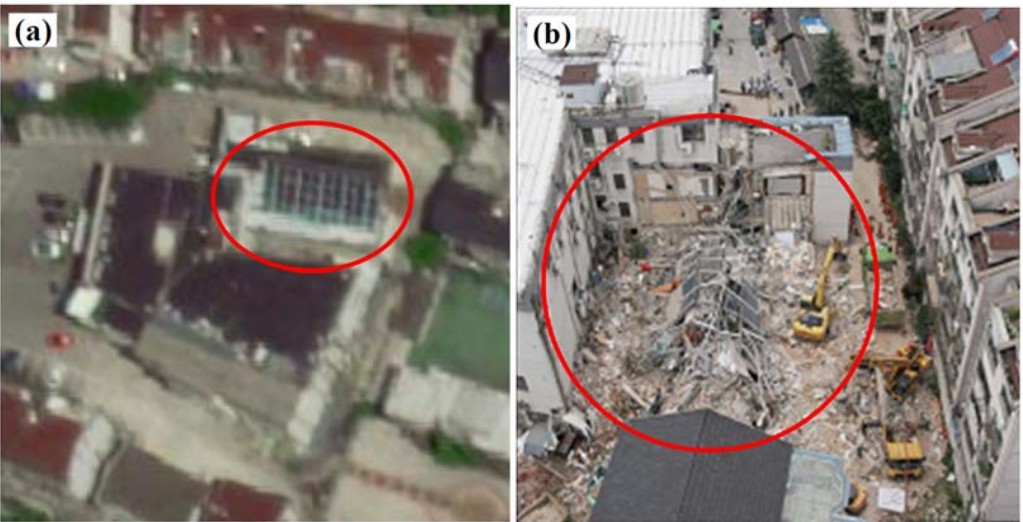

**Figure 2.** Aerial view of the auxiliary hotel building, (**a**) before and (**b**) after the breakdown (Source: https://baijiahao.baidu.com/s?id=1705141641655563947&wfr=spider&for=pc, accessed on 13 July 2021).

Preliminary investigations revealed that the main cause of the collapse was the illegal renovation of the building. The collapsed building built in the 1980s had gone through many illegal renovations throughout its life span. Particularly, the one that was carried out at the time of the collapse had not received any approval. However, the Chinese regulations stipulate that every building must undergo a thorough inspection and receive the approval of relevant authorities before rehabilitation or renovation operations. The reason is that, due to the change in ownership, building structures are generally transformed over the years, resulting in a slow accumulation of risks [22]. For example, on the surface, a single building decoration may seem harmless, while internally, it may deteriorate the bricks and concrete composing its structure. This is particularly true for the 30-year-old building investigated herein. After many instances of decoration and expansion, a fourth floor was added to the building, leading to a greatly increased load. Additionally, the internal load-bearing wall was seriously damaged in the decoration. This old auxiliary hotel building could not withstand the massive vibrations from the renovation works that were carried out. There have been numerous building collapses in recent years in China that were also caused by illegal alterations or expansions. Regardless, in case of such accidents, it is crucial to provide prompt and efficient rescue responses.

*3.2. Rescue Operations*

Figure 3 shows the firefighters and other rescue forces searching for and rescuing the victims. In fact, the Jiangsu provincial government set up an operational headquarters to efficiently manage the rescue operations, including the rescue logistics, rescue strategy and rescue sequence. Specifically, over 650 rescuers and approximately 120 rescue vehicles were

deployed on the site. No big machinery was brought to the site to prevent a secondary collapse. Additionally, the remaining structure of the building was strengthened with steel wire ropes to prevent possible collapse.

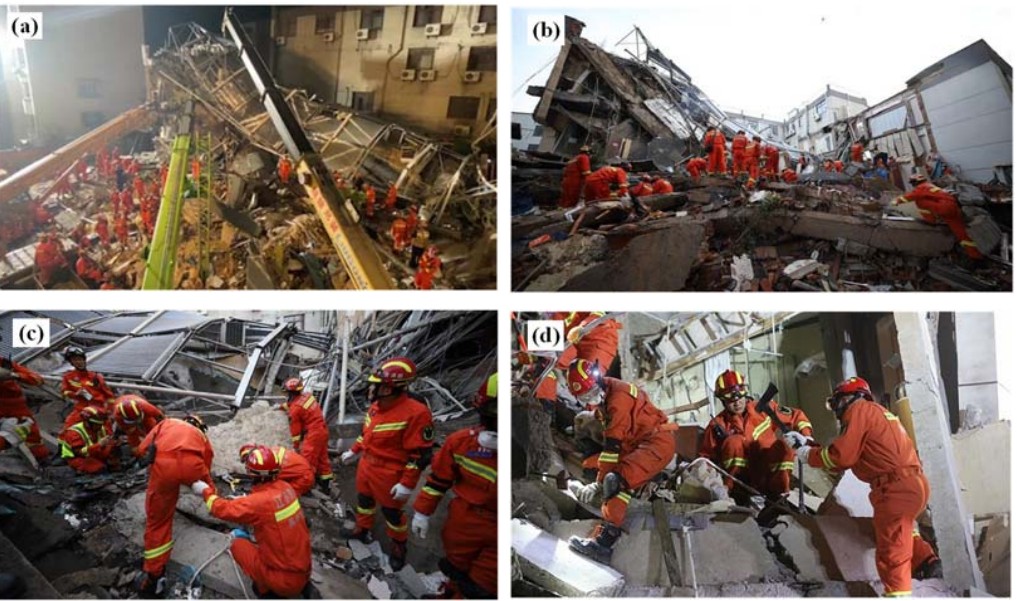

**Figure 3.** Rescue teams in action soon after the hotel collapse on 12 July 2021: (**a**) Removing collapsed building roof; (**b**) Searching lives; (**c**) Removing collapsed building materials; (**d**) Breaking large blocks of materials (Source: https://www.163.com/dy/article/GEP44IPC05503FCU.html, accessed on 13 July 2021).

It was assessed that the building had fallen down onto the ground completely and there was no possibility of a second collapse before rescue. The scene of the accident site was divided into six searching areas, while the rescuers were divided into three "24 h uninterrupted search and rescue" teams. The analyses of the hotel check-in information and authorities' investigation reports revealed that 23 people were trapped under the rubbles. Subsequently, metal cutting machines, life detectors, and sniffer dogs were deployed on the site to increase the efficiency of the entire process.

Thanks to this strategy, the number of victims were reduced to a relatively satisfactory extent. The overall rescue operations were completed within 42 h as follows: (i) at 3:00 pm on July 13 (24 h after the accident happened), 14 people were found after a thorough search. Among them, one person was unhurt, five were injured, and eight were dead. That is, 24 h later, nine people were still missing. (ii) At 9:00 am on July 14 (42 h after the accident happened), the search and rescue works ended with the discovery of all the missing victims [25]. In total, 17 people were killed, while 6 were timely rescued (among which, 1 went back home without any harm).

## 4. Discussion on the Collapse Accident

This section expands the understanding of the Suzhou's collapse accident by discussing the causes of the accident. These three aspects include the illegal renovations and extensions of the building that collapsed, the quality problem of old buildings, as well as the laxism of relevant authorities. Figure 4 shows the changes of the collapsed building to better understand the cause of collapsing accident.

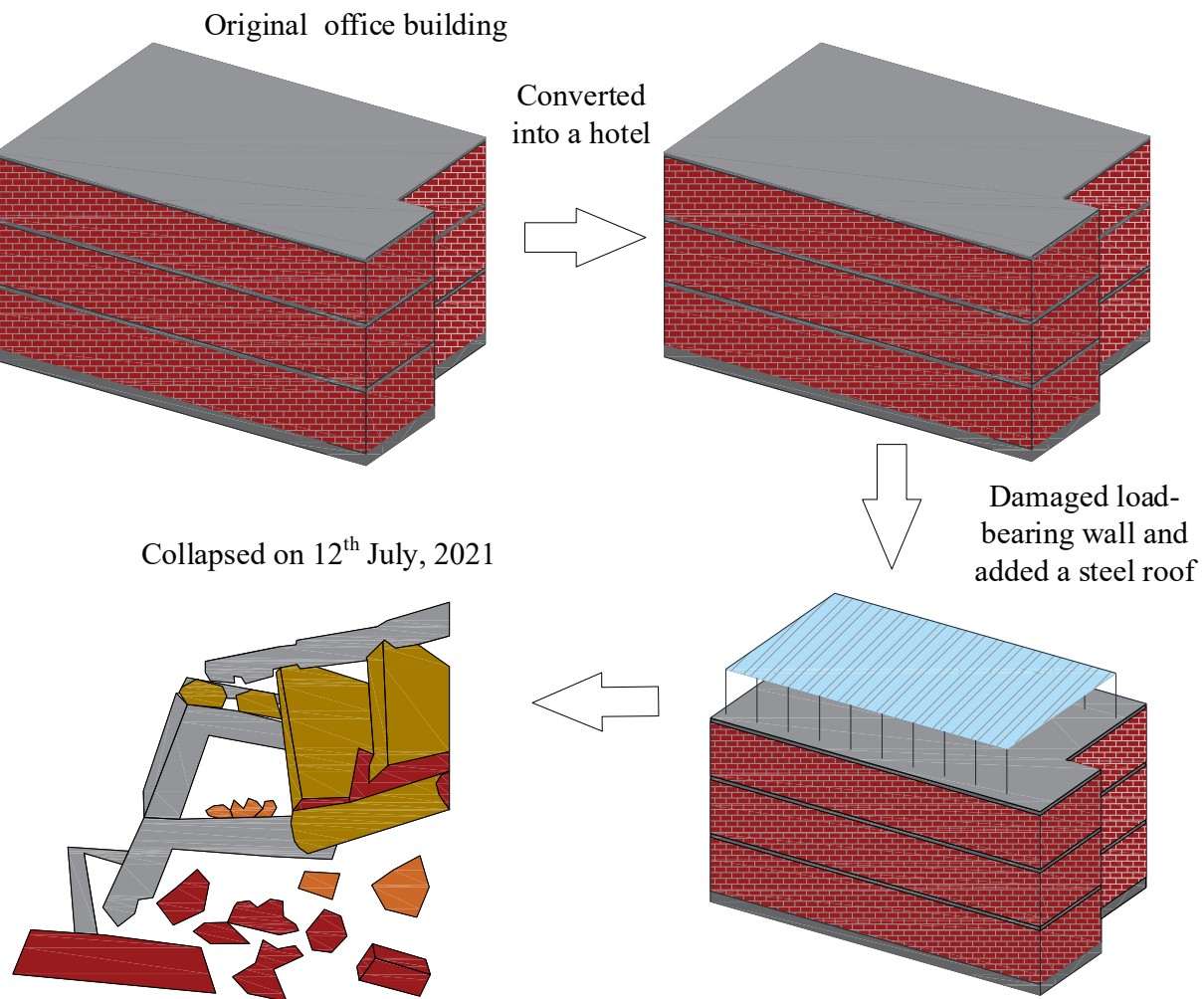

Original office building

Converted into a hotel

Damaged load-bearing wall and added a steel roof

Collapsed on 12th July, 2021

**Figure 4.** Remodeling of collapsed buildings over the last 30 years.

*4.1. Direct Cause*

4.1.1. Structural Failure

Structural mechanics analysis requires accurate building structures information. Although the construction time of the collapsed building is too early to find its building structure and inner structure, the building structure can be inferred from other information. Firstly, it can be indicated from the bricks and pre-fabricated floors in the rescue scene photo shown in Figure 5 that it is a masonry-concrete structure. Secondly, the building time of the collapsed building was 1980s, and masonry-concrete structure was the most commonly used structure in 1980s.

As shown in Figure 6, the load-bearing wall is the main loading-bearing unit in masonry-concrete structure. Different from the frame structure and shear wall structure commonly used in recent decades, the walls in masonry-concrete structure cannot be moved. Preliminary investigation show that the collapsed building was originally constructed as an office building and changed into a hotel. Over the next three decades, it had undergone numerous illegal renovations and extensions, and its interior walls that had been used for load bearing had been damaged. So, its structure has been changed, and other units bore more load for a long time. The ongoing renovation before the collapse accident also disturbed the structure, and it is the most possible cause that led to structure failure.

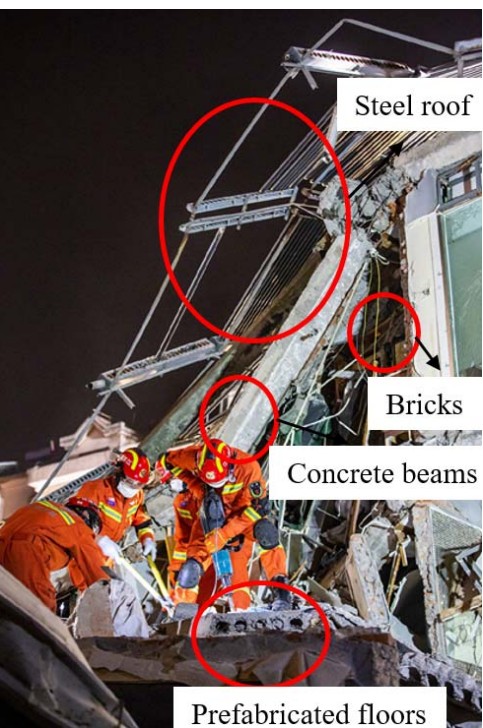

**Figure 5.** Photo of collapse site. (Picture Source: https://weibo.com/u/2803301701?topnav=1&wvr=6&topsug=1, accessed on 13 July 2021).

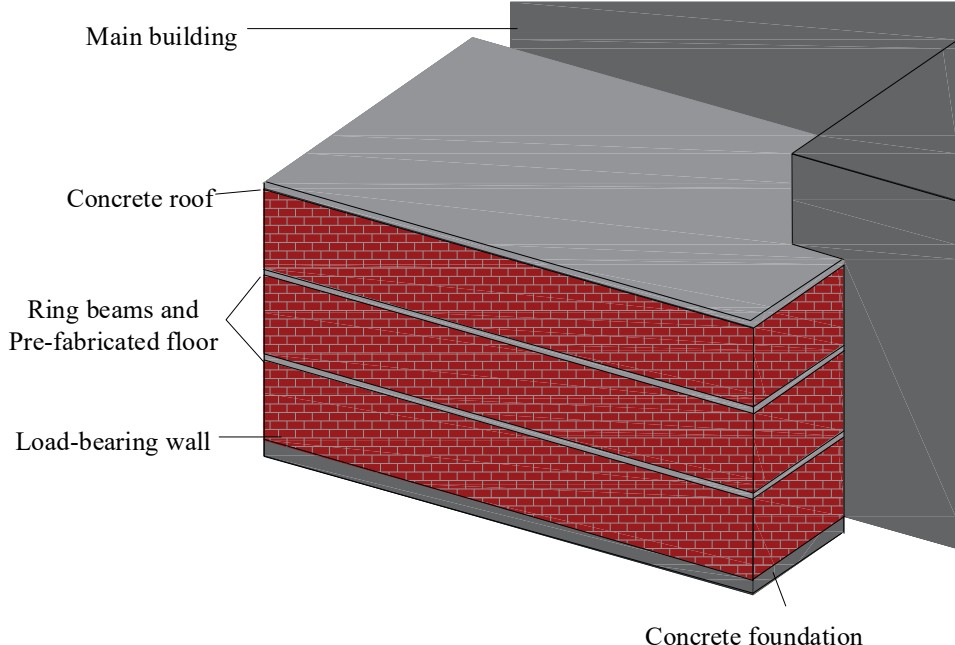

**Figure 6.** Sketch of masonry-concrete structure.

### 4.1.2. Overloading

As mentioned in Section 4.1.1, the collapsed building was built for an office building and later used for hotel. According to GB50068-2001 [26], the live load standard value of the hotel floor is greater than that of the office building. So, the load that this building was bearing must have increased and led to overloading. What is more serious is that the fourth floor was a later-added steel structure. Building a new floor would greatly increase the

load value. The building was under the condition of overloading for a long time, leading to structure fatigue. This is also an important factor leading to collapse.

### 4.1.3. Inherent Structural Defect

The investigation of previous collapse buildings in China as well as the lessons learned from the case discussed herein revealed that the masonry-concrete structures with prefabricated floors play a critical role in the breakdown process. Most of the collapsed buildings were composed of masonry-concrete structures and prefabricated floors, which are more hazardous than the frame structure or shear wall structure with cast-in-place floors. Although the masonry-concrete structures with prefabricated floors have gradually been abandoned for the benefit of the latter, there are still several buildings (e.g., the case discussed in this paper) that use this technology. It can be stipulated that the precarity (age) of a building combined with recurrent renovations will inevitably fragilize its structure and lead to its collapse.

### *4.2. Management Problem Discussion*
### 4.2.1. Problems of Old Building Management

Since the 1980s, urbanization has accelerated throughout China, with a large number of densely built buildings. However, due to the lack of efficient construction technologies and standards [27], the buildings constructed during this period are still shrouded by huge safety risks [28]. In recent years, China has improved the management of dangerous urban houses, demolishing them or repairing them. However, there are still some imperfections.

The hotel that collapsed had been built more than 30 years ago and deserved special supervision, but the local government had not conducted a systematic inspection of the building during the investigation of dangerous houses in recent years. Repeated illegal alterations and renovations to the building had also escaped the attention of relevant departments. Yet, the renovation that was under way before the breakdown was carried out without the approval of relevant departments. This new accident thus stresses the urgent necessity to strengthen the management and control of construction operations on old buildings.

### 4.2.2. Negligence of Relevant Actors

With the continuous development of China's economy, buildable lands have become so scarce that property transactions are becoming more frequent [29]. In this social context, many property owners clandestinely renovate or expand their buildings in order to make greater profits [30,31]. This also explains why many building collapses in recent years in China (Table 1) have been related to illegal renovations or extensions.

China's real estate system was established in the late 1980s. In other words, the buildings built before that period have been difficult to monitor given that many were unlicensed [30]. The latest ownership transaction of the "Four seasons Kaiyuan Hotel" dated back to 2018. Based on the property's ownership certificate, both the main and auxiliary buildings (see Figure 2) existed at the time of the transaction. Yet, according to the certificate, the "hotel buildings" (presumably referring to the main building) was a four-story structure built in 2010. Conversely, no mention was made that the building that collapsed was a three-story building built in the 1980s. All this clearly illustrates negligence in the control and supervision of property transactions. Several illegal renovations to the building were not detected or reported by the regulatory authorities, which also shows a certain negligence on their part. Again, this emphasizes the need to standardize and strictly implement the regulations and codes in place.

### *4.3. Future Remedies to Avoid Building Collapsing*
### 4.3.1. Technology Aspect

From the photos shown in Figures 3 and 4, the building is in complete collapse, and the specific failure element that led to the collapse cannot be determined. According to a

former investigation report of the collapse of a masonry-concrete structure in Shanghai [32] and Harbin [33], the collapse of the load-bearing wall caused by excessive load is the direct cause. Cracks and excessive tilting are also important signs of possible collapse [34]. Therefore, it is of vital importance to establish a monitoring system for old buildings. Load-bearing wall integrity, cracks, and tilt amount should be included.

More risk assessments are significant, especially for buildings that are more than 20 years old. In view of the current defects in the management of old (more than 20 years old) buildings, it is necessary to put forward a set of risk assessment methods for old buildings, including subjective methods [35,36] and objective methods [37–43]. These methods have been successfully used in many other areas such as environment protection, construction safety, and geological disaster prediction [44–49].

### 4.3.2. Management Aspect

To avoid recurrence of building collapse accidents, more strict management measures should be carried out. From the accident in Suzhou, this paper has the following suggestions for management:

(1) More comprehensive inspection of old buildings is required. This collapsed hotel was apparently not inspected before it collapsed, despite government requests to inspect old buildings and conduct risk assessments.

(2) Property right transactions of buildings need to be more tightly controlled. There exists serious fraud on the property right transaction certificate of this house, because it is not clear about the collapsed building in that certificate. Therefore, it is necessary to pay much more attention to the verification of building information in future property right transactions.

### 5. Conclusions

This paper reports the collapse of a hotel that took place in Suzhou City on 12 July 2021. The collapse was attributed principally to illegal renovations and extensions. The building was constructed as office building and later used as hotel. Multiple illegal renovations and extensions contain an addition floor, the removal of load-bearing walls, and the addition of isolation walls, which led to the structure's failure. Maladministration and carelessness of relevant authorities allowed illegal renovations and extensions. It was found that, at the time of the accident, the building was more than 30 years old and had undergone several illegal (structural and aesthetical) reconstructions. This situation was "encouraged" by the negligence of the relevant departments, because the failure to check the risk level of the building at the correct time greatly contributed to the accident. Based on the above, it is therefore suggested to: (i) establish and improve long-term safety management mechanisms for building renovation, decoration, and construction; (ii) standardize and enforce the rules and regulations in place—safety and risks assessments must be carried out before any building transformation; and (iii) enhance the safety awareness of residents, especially regarding cracks and excessive vibrations of buildings that are more than 20 years old.

**Author Contributions:** Conceptualization, Y.-L.C., P.G.A.N. and L.-S.Z.; formal analysis, Y.-L.C. and P.G.A.N.; investigation, Y.-L.C., P.G.A.N. and L.-S.Z.; resources, Y.-L.C. and P.G.A.N.; data curation, Y.-L.C. and P.G.A.N.; writing—original draft preparation, Y.-L.C. and P.G.A.N.; writing—review and editing, Y.-L.C., P.G.A.N. and L.-S.Z.; visualization, Y.-L.C. and P.G.A.N.; supervision, P.G.A.N. and L.-S.Z.; funding acquisition, L.-S.Z. All authors have read and agreed to the published version of the manuscript.

**Funding:** This research has been supported by the National Natural Science Foundation of China (Grant No. 42102308). and the Research Funding of Shantou University for New Faculty Member (Grant No. NTF21008-2021).

**Conflicts of Interest:** The funders had no role in the design of the study; in the collection, analyses, or interpretation of data; in the writing of the manuscript, or in the decision to publish the results.

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
