# Peer review of "A Brief Report of Hotel Collapse Causing Casualties in Suzhou, China"

_safety_

Round 1
Reviewer 1 Report
A careful review of the manuscript “A Brief Report Of Hotel Collapse Causing Casualties In Suzhou, China” has been completed. Even though the authors tried to evaluate old buildings’ condition and risks stemming from remedial operations, it is not clear what is the main objective of the manuscript and how these systems can be implemented in practice to avoid future disasters; also, it is very difficult to follow what is presented. This paper would be useful for engineers to evaluate risk level of the building. However, this investigation is not comprehensive and there are still rooms to improve. Therefore, this manuscript is not recommended for publication in Journal of Safety since paper has a critical and serious problem explained below:
- English needs to be improved. I had difficulty to follow the text and had to read the same sentence several times.
- The originality is not explained in detail.
- References are not cited sufficiently and appropriately.
Author Response
Dear Reviewer, thank you for your valuable comments. Please check the response to the comments as below.
|
Comments |
Response |
|
it is not clear what is the main objective of the manuscript |
The authors gratefully thanks for these useful comments. This communication is to present a comprehensive understanding of old building collapses accidents and supply effective recommendations. The main objective of this manuscript is clearly stated in the last paragraph of “Introduction” part, please see line 63-69 of the revised manuscript. |
|
how these systems can be implemented in practice to avoid future disasters |
The authors appreciate the comment raised by the reviewer. We talked about the possible application of these systems from two aspects, i.e., technology and management. The detailed statement about the implementation of these systems has been presented in Section 4.3. Please see line 244-270 of the revised manuscript. |
|
this investigation is not comprehensive and there are still rooms to improve. |
The authors thank the reviewer’s comments and suggestion. More detail of preliminary investigation has been added in 3.1. Please see line 95-121 of the revised manuscript. |
|
English needs to be improved. I had difficulty to follow the text and had to read the same sentence several times. |
The authors appreciate the reviewer for this comment and the English is double checked. |
|
The originality is not explained in detail. |
The authors thanks reviewer for this comment. The originality of this collapsed building is explained in Section 4.1 in the revised manuscript. We summarized the possible causes of this collapsed building and made some discussions on these causes item by item. Some new text is added and modified in line 157-208. |
|
References are not cited sufficiently and appropriately. |
The authors appreciate the comment raised by the reviewer. Some related references have been added and cited in this manuscript as references [44-49]. |

Reviewer 2 Report
The authors present a short communication for building failure. The topic is worthwhile.
However, the authors need to add:
- more literature review on the topic
- more details of structural elements and how the load actions incur
- failure analysis of the building
- recommendation of new solutions to aid the future design and construction.
For example, the authors should discuss the use of technology to improve building design and monitoring such as https://doi.org/10.3390/su11010159
Author Response
The authors would like to thank the viewer for the useful comments. Please check the response to the comments below.
|
Reviewer 2# |
more literature review on the topic |
The authors thanks for the comment raised by the reviewer. We have added more text talking about these topic in Introduction. Please see line 42-62 of the revised manuscript. Some references are added as well. |
|
more details of structural elements and how the load actions incur and failure analysis of the building |
Thank you for constructive suggestions. It is difficult to obtain the design documents of this collapsed building as it is a long time ago. Generally, this kind of building, there is no design. In order to reply this comments, we added Figure 4 to better understand the process of load actions change and a schematic for this type of masonry-concrete structure as shown in Figure 6. The mechanism of load action is similar for this type of structure. The descriptions and structural element, load actions and failure analysis are presented in the revised manuscript, please see 4.1 in line 162-208. |
|
|
recommendation of new solutions to aid the future design and construction. |
Thank you for constructive recommendation. The type of masonry-concrete structure is commonly in 1980s in China and there still exist many buildings completed in that period right now. The suggested solutions for this type of buildings are mainly health monitoring, management measures, and reinforcement design for old buildings. Specific new solutions to the future have been added in line 244-270. |

Reviewer 3 Report
The paper presents the "Four seasons Kaiyuan Hotel" collapse in the Wujiang district of Suzhou, Jiangsu Province, China.
The paper cover an interesting case and gathering experiences on past collpases is foundamental to prevent and understand future buildings failures.
However the paper is very general and does not tell any specific reason for the collapse other than geneal lack of controls and illegal renovations.
Author Response
The authors appreciate the point raised by the reviewer. Please check the response to the comments below.
|
The paper cover an interesting case and gathering experiences on past collpases is foundamental to prevent and understand future buildings failures. However the paper is very general and does not tell any specific reason for the collapse other than general lack of controls and illegal renovations. |
The authors appreciate the point raised by the reviewer. We summarized the collapse reasons item by item and discussed the reasons from different aspects, such as, structural failure, overloading, inherent structural defect, etc. We modified and extended Section 4 to provide a comprehensive explanation on the collapse reason. Figure 4 was added to better understand the process of load actions change and Figure 6 was added to show the commonly adopted building structures in the past decades in China. (The detailed text is shown in line 157-208). |
Reviewer 4 Report
Paper referred to building collapses in recent years in China. Most of them resulted from Illegal building extension or renovation. The problem of the lack of professional design and supervision is the cause of many construction collapse.
In discussion on the collapse accident the authors showed the reasons - ilegal renovations and extensions; problem of quality of old buildings; neglect of construction supervision. The authors rightly point to establish and improve long-term safety management mechanisms for building renovation, decoration and construction, standardize and enforce the rules and regulations in place and enhance the safety awareness of constructors and residents.
Attention
Professional analysis of the building statics should also be performed during a rescue operation. Failure to perform such an analysis could lead to a secondary construction collapse and more victims. I believe that the Authors should pay attention to this in their conclusions. Also, no direct reasons for the collapse are given - soil subsidence, seismic shock, corrosion processes, mechanical damage, etc. For this reason, I consider minor revision to be performed.
Author Response
The authors would like to thank the viewer for the useful comments. Please check the response to the corresponding comments below.
|
Professional analysis of the building statics should also be performed during a rescue operation. |
Thank you for your comment. The collapsed building is a masonry-concrete structure and the whole building had fallen down onto the ground. There was no possibility of a second collapse before rescue. Some text is added in line 133-134. |
|
Also, no direct reasons for the collapse are given |
Thank you for your advice, we summarized the collapse reasons item by item and discussed the reasons from different aspects, such as, structural failure, overloading, inherent structural defect, etc. We modified and extended Section 4 to provide a comprehensive explanation on the collapse reason. Figure 4 was added to better understand the process of load actions change and Figure 6 was added to show the commonly adopted building structures in the past decades in China. (The detailed text is shown in line 157-208). |
Round 2
Reviewer 1 Report
I think this still need some improvements. However, I leave further evaluation for editor.
Reviewer 2 Report
That is ok. Please accept it.Reviewer 3 Report
Accepted
Reviewer 4 Report
Paper doesn't have much scientific soundness.
Due to the importance of the problem, I propose to publish it in its present form. (With amendments made)